# A Phase I Dose Escalation and Expansion Study of Epidiolex (Cannabidiol) in Patients with Biochemically Recurrent Prostate Cancer

**DOI:** 10.3390/cancers15092505

**Published:** 2023-04-27

**Authors:** Zin W. Myint, William H. St. Clair, Stephen E. Strup, Donglin Yan, Ning Li, Derek B. Allison, Laurie E. McLouth, Carleton S. Ellis, Peng Wang, Andrew C. James, Patrick J. Hensley, Danielle E. Otto, Susanne M. Arnold, Robert S. DiPaola, Jill M. Kolesar

**Affiliations:** 1Department of Medicine, Division of Medical Oncology, University of Kentucky, Lexington, KY 40536, USA; 2Markey Cancer Center, University of Kentucky, Lexington, KY 40536, USA; 3Department of Radiation Medicine, University of Kentucky, Lexington, KY 40536, USA; 4Department of Urology, University of Kentucky, Lexington, KY 40536, USA; 5Department of Biostatistics, College of Public Health, University of Kentucky, Lexington, KY 40536, USA; 6Department of Pathology and Laboratory Medicine, University of Kentucky, Lexington, KY 40536, USA; 7Department of Behavioral Science, University of Kentucky, Lexington, KY 40536, USA; 8College of Pharmacy, University of Kentucky, Lexington, KY 40536, USA

**Keywords:** epidiolex, biochemical recurrence prostate cancer, prostate specific antigen, cannabinoid receptor 1 expression, patient reported quality of life

## Abstract

**Simple Summary:**

Cannabinoids have been widely used for pain, nausea, and appetite stimulation, and have also shown anti-tumor activity in preclinical studies of prostate cancer. Epidiolex is an oral cannabidiol solution that is FDA approved for the treatment of certain types of seizures in patients one year of age and older. We studied phase I Epidiolex dose escalation followed by dose expansion in patients with biochemically recurrent prostate cancer. A total of 21 patients were enrolled. No dose-limiting toxicities were observed at any dose level. The recommended phase 2 dose was 800 mg daily. An additional 14 patients were enrolled in the dose expansion. The most common adverse events were 55% diarrhea (grade 1–2), 25% nausea (grade 1–2), and 20% fatigue (grade 1–2). Epidiolex at a dose of 800 mg daily appears to be safe and tolerable in patients with BCR prostate cancer, supporting a safe dose for future studies.

**Abstract:**

Purpose: Cannabinoids (CBD) have anti-tumor activity against prostate cancer (PCa). Preclinical studies have demonstrated a significant decrease in prostate specific antigen (PSA) protein expression and reduced tumor growth in xenografts of LNCaP and DU-145 cells in athymic mice when treated with CBD. Over-the-counter CBD products may vary in activity without clear standardization, and Epidiolex is a standardized FDA-approved oral CBD solution for treatment of certain types of seizures. We aimed to assess the safety and preliminary anti-tumor activity of Epidiolex in patients with biochemically recurrent (BCR) PCa. Experimental design: This was an open-label, single center, phase I dose escalation study followed by a dose expansion in BCR patients after primary definitive local therapy (prostatectomy +/− salvage radiotherapy or primary definitive radiotherapy). Eligible patients were screened for urine tetrahydrocannabinol prior to enrollment. The starting dose level of Epidiolex was 600 mg by mouth once daily and escalated to 800 mg daily with the use of a Bayesian optimal interval design. All patients were treated for 90 days followed by a 10-day taper. The primary endpoints were safety and tolerability. Changes in PSA, testosterone levels, and patient-reported health-related quality of life were studied as secondary endpoints. Results: Seven patients were enrolled into the dose escalation cohort. There were no dose-limiting toxicities at the first two dose levels (600 mg and 800 mg). An additional 14 patients were enrolled at the 800 mg dose level into the dose expansion cohort. The most common adverse events were 55% diarrhea (grade 1–2), 25% nausea (grade 1–2), and 20% fatigue (grade 1–2). The mean PSA at baseline was 2.9 ng/mL. At the 12-week landmark time-point, 16 out of 18 (88%) had stable biochemical disease, one (5%) had partial biochemical response with the greatest measurable decline being 41%, and one (5%) had PSA progression. No statistically significant changes were observed in patient-reported outcomes (PROs), but PROs changed in the direction of supporting the tolerability of Epidiolex (e.g., emotional functioning improved). Conclusion: Epidiolex at a dose of 800 mg daily appears to be safe and tolerable in patients with BCR prostate cancer supporting a safe dose for future studies.

## 1. Introduction

Biochemically recurrent (BCR) prostate cancer is defined as a rising prostate specific antigen (PSA) [1] level in the absence of visible disease on imaging, such as conventional scans (CT and bone scan) and PET/CT scan (either Axumin PET/CT scan or PSMA PET/CT) [2,3]. BCR is defined as (1) PSA ≥ 0.2 ng/mL without radiographic or clinical progression after radical prostatectomy [2] and (2) an increase in PSA of at least 2 ng/mL above the post-radiation PSA nadir as per Phoenix criteria [3]. The risk factors associated with BCR are PSA doubling time, pathologic grade group, staging, and the time to PSA recurrence [4,5,6,7,8,9]. The NCCN guideline for BCR states that expectant management is also appropriate in certain patients, given the known long-term toxicity of androgen deprivation therapy (ADT) [10]. Many studies have shown that initiating early ADT therapy does not prolong overall survival or delay time to metastases [11,12]. In addition, the long-term use of ADT has been associated with many side effects including cardiovascular-related death, sleep disturbance, mood swings, sexual dysfunction, and a poorer quality of life [13,14]. Thus, there is a need for a low-toxicity option with non-hormonal therapy for men with BCR. 

Cannabinoids have been widely used in medicines for centuries to control pain, nausea, or vomiting, and to stimulate appetite, especially in cancer patients [15,16,17,18]. The *Cannabis sativa* plant contains more than 500 phytocannabinoids; tetrahydrocannabinol (THC) and cannabidiol (CBD), non-psychotropic phytocannabinoids, are among the most studied in recent years [19].

In 1974, the discovery of cannabinoids’ antagonistic effects in the male reproductive system has attracted much attention in prostate cancer research [20]. Kolodny et al. examined testosterone levels with chronic intensive marijuana use (defined as four or more days a week for a minimum of six months) [11]. The plasma testosterone level was significantly reduced by almost 50% among intensive-use marijuana participants compared to never-users [20].

Sarfaraz et al. demonstrated that both CB1 and CB2 receptors were highly expressed in cultured prostate cancer cells compared to normal prostate cell lines. Two cell lines, PrEC (normal) and LNCaP (cancer) were treated with WIN-55 212-2, a potent cannabinoid receptor agonist, for 24- and 48-h and evaluated for cell viability [21]. A dose-dependent decrease in LNCaP cell viability was seen. Results also showed a significant decrease in PSA protein expression as a dose-dependent effect of WIN-55 212-2 in LNCaP cells when the same cell lines were pretreated with SR141716 (CB1 antagonist) or SR144528 (CB2 antagonist) and subsequently administered WIN-55 212-2 [21,22]. Increased apoptosis of prostate cancer cells treated with cannabinoid agonists is a potential via G0/G1 cell cycle arrest produced by downregulation of cyclin D, cyclin-dependent kinases, and induction of p53 [23].

The effect of CBD BDS (a botanical drug substance which refers to the use of a botanical form of cannabis used as a drug) was studied using xenograft tumors of LNCaP and DU-145 cells in athymic mice. The mice were divided into six different groups and treated with CBD BDS, bicalutamide, or also co-administered docetaxel or bicalutamide at different doses [24]. The study showed that CBD BDS significantly inhibited xenograft growth, and combined CBD BDS with bicalutamide significantly prolonged survival as compared with bicalutamide or CBD BDS alone after 47 days of treatment [24].

Cannabinoids have anti-tumor activity in preclinical studies as described above, but products may vary in activity without clear standardization. Epidiolex is a standardized and FDA-approved oral CBD solution for treatment of certain types of seizures in patients 2 years of age and older [25]. The current FDA-approved dose is 5 mg/kg twice daily with an increase to 10 mg/kg twice daily allowed as the maximum dose. It is extracted from *Cannabis sativa* L. plants and contains pure CBD (>95%) with less than 0.5% THC. The known side effects with Epidiolex were diarrhea, transaminase elevations, fatigue, malaise, somnolence, decreased appetite, poor quality sleep, and sleep disorder [26]. We performed an open-label, single center, phase I/Ib study of safety and preliminary efficacy of CBD using Epidiolex in biochemically recurrent prostate cancer patients.

## 2. Methods

### 2.1. Patients

This phase 1 trial (NCT 04428203) was opened on 3 August 2020, and closed to accrual on 20 August 2021. The key eligibility criteria included men with biopsy-proven adenocarcinoma of the prostate who received definitive localized therapy with either prostatectomy or primary radiotherapy to the prostate, and now had BCR prostate cancer. BCR prostate cancer is defined as (1) PSA ≥ 0.2 ng/mL that has increased above the nadir following a radical prostatectomy, or (2) PSA 2 ng/mL above the post-therapy nadir after primary radiotherapy, or (3) PSA ≥ 0.2 ng/mL after a primary radical prostatectomy followed by salvage radiotherapy and/or radiation therapy to the oligo-metastatic bone or lymph nodes. Patients were excluded if they had any radiological evidence of metastatic disease, use of ADT within the previous three months prior to enrollment, and concurrent use of over-the-counter CBD oil, marinol, or marijuana use. All patients were pre-screened for urine THC and distress thermometer score to identify any underlying depression or suicidal ideation. Trained counselors were alerted to high distress scores to evaluate these patients and provide counselling at the point-of-care, and remained in direct contact with the care providers and team. Patients with positive urine THC were given a wash-out period of one week and allowed to participate in the study when the repeat urine THC became negative.

The full study protocol is available in Appendix A. The protocol was approved by the University of Kentucky institutional review board and the approval number is 56982. All patients provided written informed consent.

### 2.2. Study Design

In this phase 1 study, a BOIN design was used to determine the maximum tolerated dose (MTD) of Epidiolex. The initial dose level 1 was 600 mg oral solution once daily and the doses were escalated/de-escalated using the BOIN design as follows: level -2 was Epidiolex 200 mg once daily, level 1 was Epidiolex 400 mg once daily, and level 2 was Epidolex 800 mg once daily. Once MTD was defined, an additional 14 patients were treated at the MTD dose level for a total of 90 days, followed by a 7–10 days taper period for everyone on the study to evaluate long-term safety and tolerability. Patients were followed for adverse events (AEs) for a total of 30 days after the discontinuation (last dose) of Epidiolex. The total duration of long-term follow-up was 120 days. Compliance with oral Epidiolex was documented by a medication diary and medication bottles.

The primary endpoint was to evaluate the acute toxicity and long-term safety and tolerability of an Epidiolex dose in patients with BCR prostate cancer. Acute toxicity was defined as experiencing a dose-limiting toxicity (DLT) within 30 days after the initiation of study treatment. DLT was defined as grade ≥3 nausea, vomiting, diarrhea that persists >72 h despite optimal anti-emetics and anti-diarrhea treatment, grade ≥3 hematological adverse events (AEs), or grade ≥2 suicidal ideation. Long-term safety and tolerability were evaluated by AEs that occurred within 90 days after the initiation of study treatment. Secondary objectives included PSA changes (biochemical response), testosterone levels from baseline throughout the treatment period, and changes in patient-reported health-related quality of life (HRQoL), an important marker of tolerability. Exploratory objectives included assessment of CBD receptor 1 expression from prostatectomy specimens for whom archival surgical specimens were available. 

The biochemical response (PSA changes) is defined as below.

Complete Biochemical Response: Normalization of PSA (PSA level becomes undetectable) documented by two different measurements taken at least 4 weeks apart.Partial Biochemical Response: Reduction in PSA of ≥25% (from baseline) documented by two different measurements taken at least 4 weeks apart.Stable Biochemical Disease: An increase in baseline PSA of <25% documented by two different measurements taken at least 4 weeks apart.PSA Progression: An increase in baseline PSA of ≥50% documented by two different measurements taken at least 4 weeks apart.Clinical Progression: Any positive metastases findings on a bone scan and/or CT scan of the abdomen or pelvis or Axium PET/CT scan or development of symptoms attributable to cancer progression.

### 2.3. Patient-Reported Outcomes (PROs)

HRQoL is an important treatment outcome in itself, but also can serve as a PRO measure of treatment tolerability. Therefore, to further assess tolerability and inform future trial design, measures of patient-reported HRQoL were administered to patients at baseline and at the 12-week follow-up. The European Organization for Research and Treatment of Cancer Quality of Life Questionnaire Core 30 items version 3.0 (EORTC-QLQ-C30 [26]) and the EORTC-QLQ-Prostate Cancer Specific Module (PR25 [27]) were used for assessment. Items received responses according to a four-point Likert-type scale of ‘1 = not at all’, to ‘4 = very much’. Subscales were transformed linearly to range from 0 to 100, with higher scores reflecting more of the construct measured (e.g., better function or higher symptom burden). Internal consistency for the EORTC-QLQ-C30 and PR25 in this sample was good for most subscales (Cronbach’s alpha’s 0.70–0.93), with the exception of those in which there was little to no variability (e.g., Social Function alpha = 0.27; Nausea/Vomiting alpha = 0.0). 

### 2.4. Statistical Analysis

The target dose-limited toxicity (DLT) rate in this study was 30%. A patient was considered DLT evaluable if he completed at least 75% of the planned doses of Epidiolex in the first 30 days of the treatment, unless the reason was due to toxicity. The DLT rate at each dose level was calculated along with the Fisher’s exact confidence interval at 95% confidence level. AEs were summarized by descriptive statistics to evaluate long-term safety. Long-term safety and tolerability were evaluated through the summary statistics of AEs that occurred within the 90-day follow up period in patients who received any amount of the study drug. Changes in serial PSA and testosterone levels from baseline throughout the 90-day treatment period were represented by longitudinal profiles. PSA and testosterone levels were categorized and summarized by response rates with confidence intervals. Patients who were DLT evaluable, had baseline measurements, and had at least one post-baseline measurement were included in the analysis of secondary endpoints. The HRQoL assessments were analyzed for patients who received one or more doses of the recommended phase 2 dose and who completed one or more sections of the quality-of-life assessment. The HRQoL compliance rate was defined as the proportion of patients who completed the assessment among those eligible to complete it, excluding those who died or discontinued treatment. QoL assessments collected longitudinally were analyzed using appropriate linear models for repeated measurement data. The SAS statistical package (version 9.4; SAS Institute, Inc., Cary, NC, USA) was used for all data analyses. All hypothesis testing was conducted at a 5% significance level. 

## 3. Results

### 3.1. Phase I Dose Escalation 

Four patients were in dose level 1, but two were not evaluable (one patient was hospitalized with COVID-19, and one patient declined further study due to grade 1 headache). No DLTs occurred among the two patients. Subsequently, the next cohort was enrolled in dose level 2. A total of three evaluable patients were in dose level 2. Again, no DLTs occurred in these patients. Thus, the recommended phase 2 dose (RP2D) was selected as 800 mg daily. An additional 14 patients were in the dose expansion cohort, however, two patients declined further participation without finishing the study (one due to grade 2 diarrhea, and another one due to grade 1 insomnia). Among 21 patients, a total of 18 patients were included in the efficacy analysis (Figure 1).

### 3.2. Baseline Characteristics

Twenty-one patients were enrolled between July 2020 and October 2021 at the University of Kentucky, Markey Cancer Center. Baseline patient characteristics are summarized in Table 1. The median age was 69 (47–85), the median PSA at study entry was 1.65 ng/mL (0.24–16.3), and the median total testosterone at study entry was 309 (187–1166). The median time from diagnosis to study entry was 93 (5–204) months. The majority of patients were Caucasian (85.7%), followed by African American (9.5%). Primary treatment included radical prostatectomy in 15 (72%) patients and definitive radiation therapy in 5 (24%) patients. Ten (48%) subjects received radiotherapy as adjuvant or salvage treatment after radical prostatectomy. Pathologic tumor staging was summarized as: pT2N0 (40%), pT3aN0 (33%), pT1N0 (13%), and T3bN1 (6%). The majority of patients had Gleason 4 + 3 = 7 (38%) followed by Gleason 3 + 4 = 7 (28.5%), Gleason ≥ 8 (23.8%) and Gleason 3 + 3 = 6 (4.7%). PSA doubling time ranged from 3 months to 42 months; 14.3% were less than 6 months, 28.6% were between 6 and 12 months, and 52.4% were >12 months. 

### 3.3. Treatment-Related Adverse Events

Toxicities were graded according to Common Terminology Criteria for Adverse Events (CTCAE) version 5.0 [https://ctep.cancer.gov/protocoldevelopment/electronic_applications/ctc.htm#ctc_50 accessed on 25 January 2018. NIH. National Cancer Institute]. No DLTs were observed. We only included AEs which investigators thought were attributed to the study drug (possibly, probably, or definite). The most common acute and chronic treatment-related toxicities are summarized in Table 2. The acute and chronic treatment-related toxicities were defined as any treatment AEs within 30 days and within 90 days of initiating therapy, respectively. The most common acute toxicities included grade 1 or 2 diarrhea (55%), grade 1 or 2 nausea (25%), and grade 1 or 2 fatigue (20%) (Table 2). Some of the acute toxicities continued throughout the study; these include grade 1 or 2 diarrhea, grade 1 or 2 nausea, and grade 1 or 2 fatigue. Some of the minor AEs (such as abnormal LFTs and insomnia) were resolved after 30 days of therapy (Table 2). No patient required dose reductions. 

### 3.4. Preliminary Anti-Tumor Activity 

Eighteen patients (18) were evaluable for preliminary PSA response. At the 12-week landmark time-point, 16 out of 18 (88%) had stable biochemical disease, one (5%) had partial biochemical response with the greatest measurable decline being 41%, and one (5%) had PSA progression (Figure 2). Total plasma testosterone levels were monitored at baseline and during the study treatment. The mean total testosterone levels at baseline were 387 (standard deviation (SD) 266.9) (n = 18), then 377 (SD 242.6) for cycle 1 (n = 18), 324 (SD 148.3) for cycle 2 (n = 16), 485 (SD 224.7) for cycle 3 (n = 15), and 334 (SD 243.9) for one month post-study treatment.(n = 10). Testosterone levels were similar over the course of treatment. 

Three patients had clinical progression as per convention scans (CT CAP and bone scan); one developed oligo-metastasis disease, two progressed after the study period, and one patient died from a non-treatment or disease-related cause.

### 3.5. Preliminary Patient-Reported Outcomes

Of the 17 patients who received the recommended dose of 800 mg daily, 17 completed patient-reported outcomes at baseline and 12 completed PROs at 12 weeks.

Descriptive statistics for QLQ-C30 and QLQ-PR25 scores at baseline and at 12 weeks (post-treatment), and mean change scores for each subscale are shown in Table 3. No statistically significant changes were observed from baseline to 12-week follow-up. Non-significant improvements were observed in global HRQoL (M change = 2.9; 95% CI −8.0; 13.8), emotional functioning (M change = 2.4; 95% CI −3.3; 8.1), sexual functioning (M change = 20.9; 95% CI = 1.5; 40.3), pain (M change = −1.8; 95% CI = −16.6; 13.0), and insomnia (M change = −5.4; 95% CI = −20.3; 9.5). Non-significant worsening was observed in physical functioning (M change = −2.9; 95% CI = −13.3; 7.5), role functioning (M change = −4; 95% CI = −13.0; 5.0), cognitive functioning (M change = 01.4; 95% CI −9.8; 7.0), and social functioning (M change = −1.3; 95% CI = −8.9; 6.3). Symptoms with the most worsening included constipation (M change = 10.0; 95% = −1.6; 21.6), dyspnea (M change = 6.1; 95% CI = −9.0; 21.2), and fatigue (M change = 4.3; 95% CI = −8.5; 17.1).

### 3.6. Biomarker Assessment CB1 Receptor Expression Level by Immunohistochemistry

#### 3.6.1. Background

Sarfaraz et al. demonstrated that both CB1 and CB2 receptors were highly expressed in cultured prostate cancer cells compared to normal prostate cell lines [21]. Although immunohistochemistry (IHC) can be reliably performed to assess the expression of CB1 in formalin-fixed, paraffin-embedded tissue (FFPE), multiple clones of CB2 have been shown to be unreliable and inconsistent [28,29]. As a result, we aimed to explore the correlation between CB1 expression in available prostate cancer archival FFPE and PSA outcomes in this patient cohort. 

#### 3.6.2. Method

FFPE was sectioned at 4 microns and baked at 58 °C for a minimum of one hour. Staining was conducted with the Ventana Discovery Ultra using Standard CC2 antigen retrieval (Roche, Tucson, AZ, USA) and was incubated with an anti-CB1 antibody (ab23703, abcam, Cambridge, MA, USA) at 1:1000 for 1 h at 37 °C prior to incubation with OmniMap anti-Rabbit-HRP (Roche, Tucson, AZ, USA) and DAB (Roche), according to the manufacturer’s recommendations. Prior to permanent mounting, the slides were lightly counterstained in Mayer’s hematoxylin. CB1 receptor expression was analyzed by IHC using FFPE tissue from radical prostatectomy samples (n = 9) or core needle biopsy (n = 2), based on the availability of tissue. A CB1 receptor expression score was calculated by analyzing the intensity (scored 0 = none, 1 = weak, 2 = moderate, 3 = strong) multiplied by the quantity of tumor (1 = 1–25%, 2 = 26–50%, 3 = 50–75%, 4 => 76%) showing cytoplasmic and/or membranous expression. Based on the distribution of scores, cases were categorized as a high (≥6) vs. low (≤5).

#### 3.6.3. Result

Eleven out of 15 patients had available archival tissues. Four patients had high expression [Appendix A] (range 6–9) and 7 patients had low expression (range 1–4) [Appendix A]. The CB1 expression level was analyzed and correlated in relation to PSA changes and outcomes. There was no statistically significant correlation between the CB1 expression level and PSA changes *p* = 0.17. [Appendix A] However, it was interesting that one patient was noted to have the lowest PSA changes correlated with the highest IHC score of 9.

## 4. Discussion

This study is the first to evaluate the safety and preliminary anti-tumor activity of Epidiolex in patients with BCR prostate cancer. Findings from this open-label phase I study demonstrate that Epidiolex at a RP2D dose of 800 mg daily appears to be safe and tolerable. The most common treatment related AEs were grade 1 or 2 diarrhea, followed by grade 1 or 2 nausea, followed by grade 1 or 2 fatigue, which reflect the known Epidiolex-related AEs. No patients required dose reductions. There were no observed DLTs with Epidiolex. Reported symptoms were managed with anti-diarrhea or anti-nausea medications. The fatigue was mild to moderate and manageable for most patients, however, one patient self-withdrew from the study due to grade 2 fatigue. 

A particular concern associated with the use of Epidiolex is mental depression or suicidal ideation. There have been reports of worsening depression and suicidal ideation with the use of CBD [30,31]. In this trial, we pre-screened with distress thermometer scores to identify any underlying depression or suicidal ideation prior to enrollment, and during the treatment period. We did not observe development of depression or worsening of depression during the 120 days of monitoring. 

In addition to safety, we were also interested in patient-reported overall quality of life, emotional functioning, fatigue, appetite, sleep, and diarrhea as one of our secondary objectives. While no statistically significant changes were observed, PROs changed in the direction of supporting the tolerability of Epidiolex. Overall HRQoL and emotional function increased from baseline to 12 weeks, whereas insomnia decreased. Patients did not report appetite loss, and worsening in diarrhea was well below minimally important difference thresholds. Given the small sample size, these results should be interpreted with caution, but they provide preliminary data on the use of PROs to investigate tolerability in future trials of Epidiolex. 

In the present study, treatment at the 12-week landmark time-point with Epidiolex demonstrated the 16 out of 18 (88%) had stable biochemical disease, one (5%) had partial biochemical response with the greatest measurable decline being 41%, and one (5%) had PSA and clinical progression during the treatment period per the protocol definition described under the study design section. Most patients in the study had stable biochemical disease, and it is difficult to explain whether this is related to Epidiolex vs. underlying indolent behavior of prostate cancer. There were no significant changes with testosterone levels, suggesting Epidiolex might not affect hormonal suppression, as expected based on the proposed anticancer mechanism of Epidiolex. A larger sample size with a longer duration of treatment will be needed to test the anti-tumor efficacy of Epidiolex in prostate cancer. 

The mechanism of anti-tumor activity of plant-derived cannabinoids is complex, and the exact mechanism of Epidiolex on prostate cancer cells is unclear. It is hypothesized that plant-derived cannabinoids can activate both the cannabinoid (CB) receptor and CB receptor-independent signal transduction pathways and, subsequently, inhibit cell proliferation, induce cell cycle arrest, and cell death. The two known CB receptors, CB1 and CB2, activate through G-protein-coupled receptors by inhibiting adenylate cyclase with a resultant decrease in cAMP levels [23]. The activated CB receptors also stimulate multiple signaling pathways through non-G protein pathways such as extracellular-signal-regulated kinase 1/2 (ERK1/2), Ca2+/calmodulin-dependent protein kinase kinase (CaMKK), phosphatidylinositol 3-kinase (PI3K/Akt) [32], Ceramide, and reactive oxygen species, all of which result in apoptosis and cell cycle arrest [32,33,34,35,36,37,38]. Other proposed mechanisms using CB receptor-independent signal transduction pathways include GPR55, TRPV1, TRPV2, and TRPMB [32,33,34,35,36,37,38].

Cannabinoids have central (CB1) and peripheral (CB2) receptors. CB1 is mainly active in the brain, lungs, and reproductive organs, whereas CB2 is in the immune system and the bones [39]. The CB1 or CB2 receptor levels are highly expressed in some cancer types such as breast cancer [40] and glioma [41]. The overexpression of the CB2 receptor in Her 2+ breast cancer was shown to correlate with tumor aggressiveness and poor prognosis [42]. Although CB1 receptors were highly expressed in prostate cancer cells, there was no significant difference between CB1 expression levels and PSA changes in this small population. The CB1 expression levels did not support their use as a predictive biomarker in this study, however, additional study of this issue in a larger population may be warranted.

To our knowledge, this study represents the first clinical trial to show the safety of Epidiolex in BCR prostate cancer after definitive local therapy. Further studies are needed to confirm the biochemical activity and clinical benefit of Epidiolex in this population. There are inherent limitations to this study. The applicability of these findings is limited by a short duration of treatment with Epidiolex, a short duration of follow up, a small sample size, the open-label study design, the lack of a comparator group, and the cohort of patients with available matched tissue for CB1 expression was limited. 

## 5. Conclusions

Epidiolex at a dose of 800 mg daily was well tolerated with an acceptable safety profile in patients with BCR prostate cancer.

## Figures and Tables

**Figure 1 cancers-15-02505-f001:**
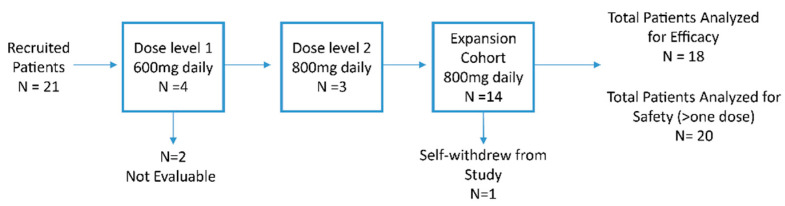
Consort Diagram. N, number.

**Figure 2 cancers-15-02505-f002:**
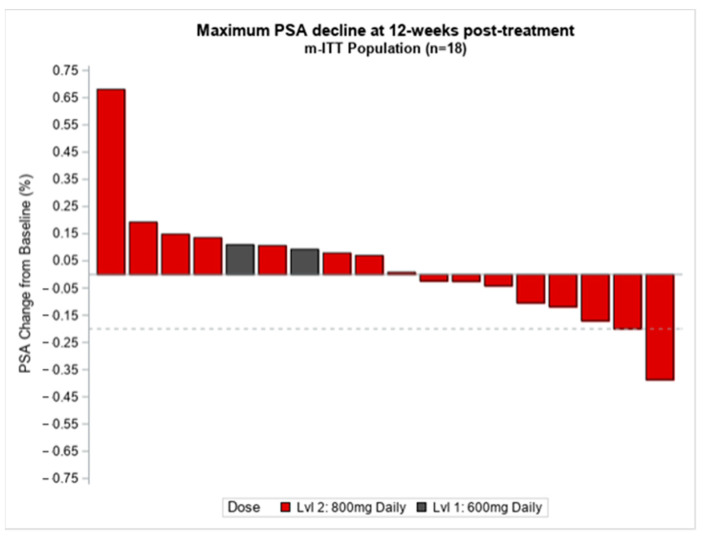
Change in PSA relative to Epidiolex treatment. PSA, prostate specific antigen, %, percentage, ITT, intention to treat, n, number, Lvl, level.

**Table 1 cancers-15-02505-t001:** Demographic and Baseline Characteristics (N = 21).

Variables	Median (Range)
Age (years)	69 (47–85)
PSA (ng/mL)	1.25 (0.24–16.3)
Testosterone, total (ng/dL)	325 (7–1166)
Time from diagnosis to study entry (months)	93 (5–204)
Age Group	Number (%)
40–49	1 (4.7%)
50–59	1 (4.7%)
60–69	9 (42.8%)
70–79	7 (33%)
80–89	3 (14.2%)
Total	21 (100%)
Race	Number (%)
White	17 (85.7%)
African American	2 (9.5%)
Asian	1 (4.7%)
Unknown	1(4.7%)
Total	21 (100%)
Prior Treatment	Number (%)
Prostatectomy Alone	4 (19.1%)
Radiation Alone	5 (23.8%)
Prostatectomy and salvage radiation	12 (57.1%)
Untreated	0 (0%)
Total	21 (100%)
Gleason grade at diagnosis	Number (%)
3 + 3 = 6	1 (4.7%)
3 + 4 = 7	6 (28.5%)
4 + 3 = 7	8 (38%)
≥8	5 (23.8%)
Unknown	1 (4.7%)
Total	21 (100%)
Staging groups for prostatectomy patients	Number (%)
T1N0	2 (13.3%)
T2N0	6 (40%)
T3aN0	5 (33.3%)
T3bN1	1 (6.6%)
Unknown	1 (6.6%)
Total	15 (100%)
PSA doubling time	Number (%)
Less than 6 months	3 (14.3%)
6 to 12 months	6 (28.6%)
>12 months	11 (52.4%)

N = number, PSA = prostate specific antigen.

**Table 2 cancers-15-02505-t002:** Treatment-Related Toxicity.

Body System	Symptom	All AEs N (%)	CTCAE Toxicity Grade					Acute Onset (within 30 Days) N (%)	CTCAE Toxicity Grade					Chronic Onset (within 90 Days)	CTCAE Toxicity Grade				
			1	2	3	4	5		1	2	3	4	5		1	2	3	4	5
Gastrointestinal	Diarrhea	11 (55%)	10	1	0	0	0	8 (40%)	7	1	0	0	0	10 (50%)	9	1	0	0	0
	Nausea	5 (25%)	5	0	0	0	0	3 (15%)	3	0	0	0	0	4 (20%)	4	0	0	0	0
	Bloating	1 (5%)	1	0	0	0	0	0						1 (5%)	1	0	0	0	0
	Stomach pain	1 (5%)	1	0	0	0	0	0						1 (5%)	1	0	0	0	0
	Abnormal LFTs	1 (5%)	1	0	0	0	0	1 (5%)	1	0	0	0	0	0					
General	Fatigue	4 (20%)	3	1	0	0	0	2 (10%)	2	0	0	0	0	4 (20%)	3	1	0	0	0
	Malaise	1 (5%)	1	0	0	0	0	1 (5%)	1	0	0	0	0	1 (5%)	1	0	0	0	0
Nervous system	Headache	2 (10%)	2	0	0	0	0	2 (10%)	2	0	0	0	0	2 (10%)	2	0	0	0	0
	Insomnia	1 (5%)	1	0	0	0	0	1 (5%)	1	0	0	0	0	0					
Skin	Rash	1 (5%)	1	0	0	0	0	0						1 (5%)	1	0	0	0	0

N = number, AE = adverse event; CTCAE = Common Terminology Criteria for Adverse Events, LFTs = Liver function test.

**Table 3 cancers-15-02505-t003:** Health-related quality-of-life assessment at baseline (prior to treatment) and at 12-weeks post-treatment using EORTC QLQ-C30 and QLQ-PR 25.

	Baseline			12-Week Post-Baseline			Mean Change	95% CI		
	N	Mean	Std	N	Mean	Std		Lower Bound	Upper Bound	*p*-Value
Global health status/QoL	17	78.4	15.0	12	81.3	13.4	2.9	−8.0	13.8	0.60
Physical functioning	17	92.9	11.9	12	90.0	14.4	−2.9	−13.3	7.5	0.56
Role functioning	17	97.1	8.8	12	93.1	13.2	−4	−13.0	5.0	0.34
Emotional functioning	17	94.1	9.2	12	96.5	5.6	2.4	−3.3	8.1	0.43
Cognitive functioning	17	93.1	10.3	12	91.7	11.2	−1.4	−9.8	7.0	0.73
Social functioning	17	97.1	8.8	12	95.8	10.4	−1.3	−8.9	6.3	0.72
Symptom Scales										
Fatigue	17	12.4	13.0	12	16.7	18.7	4.3	−8.5	17.1	0.47
Nausea/Vomiting	17	1.0	4.0	12	1.4	4.8	0.4	−3.1	3.9	0.81
Pain	17	15.7	21.6	12	13.9	17.2	−1.8	−16.6	13.0	0.81
Dyspnea	17	7.8	14.6	12	13.9	22.3	6.1	−9.0	21.2	0.38
Insomnia	17	13.7	16.9	12	8.3	20.7	−5.4	−20.3	9.5	0.45
Appetite loss	17	0.0	0.0	12	0.0	0.0	N/A	N/A	N/A	N/A
Constipation	17	3.9	11.1	12	13.9	17.2	10	−1.6	21.6	0.07
Diarrhea	17	7.8	14.6	12	8.3	15.1	0.5	−11.0	12.0	0.93
Financial problems	17	7.8	14.6	12	11.1	16.4	3.3	−8.8	15.4	0.57
Prostate−Specific										
Urinary symptoms	17	17.9	12.7	12	17.5	11.2	−0.4	−9.6	8.8	0.93
Incontinence aid	7	9.5	16.3	3	22.2	19.2	12.7	−13.3	38.7	0.29
Bowel symptoms	17	6.4	8.1	12	7.6	8.3	1.2	−5.2	7.6	0.70
Hormonal treatment−related symptoms	17	5.9	6.0	12	9.0	10.2	3.1	−3.6	9.8	0.31
Sexual activity	17	67.6	22.3	12	63.9	21.1	−3.7	−20.4	13.0	0.66
Sexual functioning	8	56.9	26.8	2	77.8	0.0	20.9	1.5	40.3	0.30

Qol, quality of life; std, standard deviation; CI, confidence interval.

## Data Availability

The raw datasets used or analyzed for the study are available from the corresponding author on reasonable request.

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
