# Peer review of "A Phase I Dose Escalation and Expansion Study of Epidiolex (Cannabidiol) in Patients with Biochemically Recurrent Prostate Cancer"

_cancers, 2023, doi:10.3390/cancers15092505_

Round 1

Reviewer 1 Report

In the introduction there is no reference to PSA which is the marker of biochemical relapse and is used for risk startification.

Ferraro, S.; Bussetti, M.;Bassani, N.; Rossi, R.S.; Incarbone,G.P.; Bianchi, F.; Maggioni, M.; Runza,L.; Ceriotti, F.; Panteghini, M. Definition of Outcome-Based Prostate-Specific Antigen (PSA) Thresholds for Advanced Prostate Cancer Risk Prediction. Cancers 2021, 13, 3381

The conclusions are not supported by the biochimical decrease of PSA and the risk of the therapy seems to be higher than the benefit.

Author Response

Reviewer 1

In the introduction there is no reference to PSA which is the marker of biochemical relapse and is used for risk startification.

Ferraro, S.; Bussetti, M.;Bassani, N.; Rossi, R.S.; Incarbone,G.P.; Bianchi, F.; Maggioni, M.; Runza,L.; Ceriotti, F.; Panteghini, M. Definition of Outcome-Based Prostate-Specific Antigen (PSA) Thresholds for Advanced Prostate Cancer Risk Prediction. Cancers 2021, 13, 3381

Authors’ response: we added the above citation to the manuscript.

The conclusions are not supported by the biochimical decrease of PSA and the risk of the therapy seems to be higher than the benefit.

Authors’ response: Our study is a phase I design and from this present study, no grade 3 toxicity was observed. Most side effects were grade 1 and some grade 2 but these side effects were easily controlled with anti-diarrhea and anti-nausea. Our primary objective of this study is safety, tolerability and dose finding. Based on the findings, we conclude Epidiolex is safe and tolerability with recommended phase 2 dose was 800mg daily. This study is not designed to answer biochemical responses. A larger study will be needed to address biochemical responses of prostate cancer.

Reviewer 2

Myint and colleagues present a nicely designed phase 1 clinical trial with an investigational agent of significant interest in this patient population given the ubiquitous use of CBD and other cannabinoid products in the oncologic community. 

Several comments for the authors follow below.

Regarding the methods, I note that the full protocol is not available for review along with the Figures presented in the supplement. 

Authors’ response: The full protocol is now uploaded.

In lines 118-119 on page 3, more explicit clarification of the definition of "long term follow up" would be helpful-- the total treatment duration is 90 days, with 30 days of additional post-treatment follow up. Does 120 days constitute the duration of long term follow up, or is there additional subsequent follow up?

Authors’ response: The total duration of long-term follow up was 120 days. No additional subsequent follow up.

Lines 180ff on page 4 regarding description of the baseline patient characteristics -- is data regarding the PSA doubling time available? This would be of interest as it is the patients with a shorter doubling time (i.e. usually < 6 months) who may be offered systemic therapy such as ADT, while patients with a slow doubling time may be more likely offered observation. It would be helpful to know the range of PSADT in this study population.

Authors’ response: PSADT data has been added in the baseline characteristics table.

Regarding interim restaging scans (line 222-224, page 7) -- what types of imaging modalities were used, CT C/A/P and bone scan or PSMA PET scan?

Authors’ response: It was with CT CAP and bone scan.  

Reviewer 3

I read with interest your manuscript on this interesting topic. 

Trying to find new therapies in the setting of BCR Pca patients it’s for sure stimulating, even If difficult. Overall, the study was well conducted, and the results are well described. 

My main concern is that the discussion should be better organized. 

It wasn’t demonstrated a strong correlation between the CB1 expression and the PSA decrease, and the number of patients who had disease progression was like that of those who experienced stable or decreased PSA levels. 

These are major points that should be more emphasized in the discussion and should be treated firstly with respect to the side effects or QoL outcomes and the safety profile because before understanding if a drug is safe we need to know if a drug is useful. 

So, what is the point of using Epidiolex? Should that be used to ameliorate emotional functioning?

Author’s response: Authors have re-arranged the discussion points as per suggestion. The treatment for BCR is controversial and due to long term ADT toxicities, studies are needed to find non-hormonal therapy or other treatment options for BCR patients. Preclinical study has shown that CBD has anti-tumor activity in prostate cancer cell lines and vivo, authors evaluated the safety and tolerability of CBD (Epidiolex) in prostate cancer population. Once we know if the drug is safe, the next study design would be to test for the efficacy.

It is a great point that Epidiolex should be used to ameliorate emotional functioning. As our study population is biochemical recurrence population and not a stage IV population, those patients generally tend to have a good baseline quality of life and emotional functioning.

Additionally, the authors should introduce a limitation paragraph before the conclusions, clearly addressing the short FU, the small number of patients involved in the study, and the possibility that the presence of other influencing factors apart from the use of Epidiolex on the PSA trend.

Authors’ response: Authors have added limitation of the study before the conclusion paragraph.

In view of the above, the conclusions paragraph should be massaged.

Consider citing these papers:

- https://pubmed.ncbi.nlm.nih.gov/31893344/

- https://pubmed.ncbi.nlm.nih.gov/31579118/

Authors’ response: Authors have added above citations.

Minor

Lines 100-103: authors should state the reference classification used to define BCR.

Authors’ response: Authors have added above citation.

Reviewer 2 Report

Myint and colleagues present a nicely designed phase 1 clinical trial with an investigational agent of significant interest in this patient population given the ubiquitous use of CBD and other cannabinoid products in the oncologic community. 

Several comments for the authors follow below.

Regarding the methods, I note that the full protocol is not available for review along with the Figures presented in the supplement. 

In lines 118-119 on page 3, more explicit clarification of the definition of "long term follow up" would be helpful-- the total treatment duration is 90 days, with 30 days of additional post-treatment follow up. Does 120 days constitute the duration of long term follow up, or is there additional subsequent follow up?

Lines 180ff on page 4 regarding description of the baseline patient characteristics -- is data regarding the PSA doubling time available? This would be of interest as it is the patients with a shorter doubling time (i.e. usually < 6 months) who may be offered systemic therapy such as ADT, while patients with a slow doubling time may be more likely offered observation. It would be helpful to know the range of PSADT in this study population.

Regarding interim restaging scans (line 222-224, page 7) -- what types of imaging modalities were used, CT C/A/P and bone scan or PSMA PET scan?

Author Response

Myint and colleagues present a nicely designed phase 1 clinical trial with an investigational agent of significant interest in this patient population given the ubiquitous use of CBD and other cannabinoid products in the oncologic community. 

Several comments for the authors follow below.

Regarding the methods, I note that the full protocol is not available for review along with the Figures presented in the supplement. 

Authors’ response: The full protocol is now uploaded.

In lines 118-119 on page 3, more explicit clarification of the definition of "long term follow up" would be helpful-- the total treatment duration is 90 days, with 30 days of additional post-treatment follow up. Does 120 days constitute the duration of long term follow up, or is there additional subsequent follow up?

Authors’ response: The total duration of long-term follow up was 120 days. No additional subsequent follow up.

Lines 180ff on page 4 regarding description of the baseline patient characteristics -- is data regarding the PSA doubling time available? This would be of interest as it is the patients with a shorter doubling time (i.e. usually < 6 months) who may be offered systemic therapy such as ADT, while patients with a slow doubling time may be more likely offered observation. It would be helpful to know the range of PSADT in this study population.

Authors’ response: PSADT data has been added in the baseline characteristics table.

Regarding interim restaging scans (line 222-224, page 7) -- what types of imaging modalities were used, CT C/A/P and bone scan or PSMA PET scan?

Authors’ response: It was with CT CAP and bone scan.  

Reviewer 3 Report

I read with interest your manuscript on this interesting topic. 

Trying to find new therapies in the setting of BCR Pca patients it’s for sure stimulating, even If difficult. Overall, the study was well conducted, and the results are well described. 

My main concern is that the discussion should be better organized. 

It wasn’t demonstrated a strong correlation between the CB1 expression and the PSA decrease, and the number of patients who had disease progression was like that of those who experienced stable or decreased PSA levels. 

These are major points that should be more emphasized in the discussion and should be treated firstly with respect to the side effects or QoL outcomes and the safety profile because before understanding if a drug is safe we need to know if a drug is useful. 

So, what is the point of using Epidiolex? Should that be used to ameliorate emotional functioning?

Additionally, the authors should introduce a limitation paragraph before the conclusions, clearly addressing the short FU, the small number of patients involved in the study, and the possibility that the presence of other influencing factors apart from the use of Epidiolex on the PSA trend.

In view of the above, the conclusions paragraph should be massaged.

Consider citing these papers:

- https://pubmed.ncbi.nlm.nih.gov/31893344/

- https://pubmed.ncbi.nlm.nih.gov/31579118/

Minor

Lines 100-103: authors should state the reference classification used to define BCR.

Author Response

I read with interest your manuscript on this interesting topic. 

Trying to find new therapies in the setting of BCR Pca patients it’s for sure stimulating, even If difficult. Overall, the study was well conducted, and the results are well described. 

My main concern is that the discussion should be better organized. 

It wasn’t demonstrated a strong correlation between the CB1 expression and the PSA decrease, and the number of patients who had disease progression was like that of those who experienced stable or decreased PSA levels. 

These are major points that should be more emphasized in the discussion and should be treated firstly with respect to the side effects or QoL outcomes and the safety profile because before understanding if a drug is safe we need to know if a drug is useful. 

So, what is the point of using Epidiolex? Should that be used to ameliorate emotional functioning?

Author’s response: Authors have re-arranged the discussion points as per suggestion. The treatment for BCR is controversial and due to long term ADT toxicities, studies are needed to find non-hormonal therapy or other treatment options for BCR patients. Preclinical study has shown that CBD has anti-tumor activity in prostate cancer cell lines and vivo, authors evaluated the safety and tolerability of CBD (Epidiolex) in prostate cancer population. Once we know if the drug is safe, the next study design would be to test for the efficacy.

It is a great point that Epidiolex should be used to ameliorate emotional functioning. As our study population is biochemical recurrence population and not a stage IV population, those patients generally tend to have a good baseline quality of life and emotional functioning.

Additionally, the authors should introduce a limitation paragraph before the conclusions, clearly addressing the short FU, the small number of patients involved in the study, and the possibility that the presence of other influencing factors apart from the use of Epidiolex on the PSA trend.

Authors’ response: Authors have added limitation of the study before the conclusion paragraph.

In view of the above, the conclusions paragraph should be massaged.

Consider citing these papers:

- https://pubmed.ncbi.nlm.nih.gov/31893344/

- https://pubmed.ncbi.nlm.nih.gov/31579118/

Authors’ response: Authors have added above citations.

Minor

Lines 100-103: authors should state the reference classification used to define BCR.

Authors’ response: Authors have added above citation.

Round 2

Reviewer 1 Report

no further comments